# Bio-Banding in Judo: The Mediation Role of Anthropometric Variables on the Maturation Effect

**DOI:** 10.3390/ijerph17010361

**Published:** 2020-01-05

**Authors:** Bruno B. Giudicelli, Leonardo G. O. Luz, Mustafa Sogut, Alain G. Massart, Arnaldo C. Júnior, António J. Figueiredo

**Affiliations:** 1Kinanthropometry, Physical Activity and Health Promotion Laboratory (LACAPS), Campus Arapiraca, Federal University of Alagoas, Arapiraca 57309-005, Brazil; 2Research Unit for Sport and Physical Activity (CIDAF), Faculty of Sport Science and Physical Education, University of Coimbra, Coimbra 3004-531, Portugal; 3Faculty of Sport Sciences, Kırıkkale University, Kırıkkale 71450, Turkey

**Keywords:** biological maturation, bio-banding, judo, combat sports, rapid weight loss, mediation analysis

## Abstract

Young judo athletes are bio-banding based on age and body mass and compete in weight classes. The purposes of this study were to investigate the influences of maturation on physical performance in young judokas through controlling the chronological age and body mass, and to examine the mediating role of anthropometric variables. Sixty-seven judokas, aged 11.0–14.7, were measured for 11 anthropometric and seven physical performance variables. Pearson partial correlations were conducted to verify the relationship between the maturational indicator and the dependent variables. Mediation analyses were performed to identify the extent to which anthropometric variables mediate the relationship. The maturation effect remained on the aerobic capacity and handgrip strength (*p* < 0.05). Fat mass (*b* = 80.335, 95%CI 11.580–204.270) and fat-free mass (*b* = 108.256, 95%CI 39.508–207.606) totally mediated the effect on aerobic capacity. Fat mass (*b* = 0.023, 95%CI 0.004–0.057), fat-free mass (*b* = 0.029, 95%CI 0.011–0.058), stature (*b* = 0.031, 95%CI 0.008–0.061), arm span (*b* = 0.021, 95%CI 0.003–0.044), and inferior members length (*b* = 0.022, 95%CI 0.005–0.049) totally mediated the effect on handgrip strength. The effect of biological maturation is noticeable even after age and body mass control, being mediated by anthropometric variables related to body composition and size.

## 1. Introduction

Chronological age is traditionally used in youth sports for the purpose of matching competitors or teams [1]. However, this approach has various limitations that at any given age there can be large maturity-associated variations in size and functional capacities among children [2,3,4,5]. Being advance in maturity may provide an advantage for young male athletes in sports characterized by power, strength, and speed [6]. In girls, it is associated with greater size and strength [7,8,9]. Therefore, several maturity-based classification methods have been proposed in the context of training and competition [10,11,12]. The latest attempt to advocate the biological maturity-based matching is denominated bio-banding. Bio-banding refers to the process of grouping young athletes into bands according to attributes related to growth and maturation status rather than chronological age [13]. The predicted mature status (PMS), a noninvasive method to estimate somatic maturity [14], is the most current method employed to assess maturity. Studies and implementations of bio-banding, on the other hand, are limited to the training or unofficial tournaments of young male soccer players [15,16,17].

The concern of grouping young athletes on the physical attributes is common in combat sports (e.g., boxing, judo, taekwondo, and wrestling) where young athletes are matched based on age and body mass and compete in a series of weight classes. This form of bio-banding classification could facilitate fair competition and reduce potential injuries [18,19]. However, doubts may be raised about the suitability of body mass-based classification to guarantee fair play in combat sports due to evidence of maturation effect within weight categories in young combat sports [20], although there is an important gap in studies on this subject, and to the widespread adoption of rapid weight loss (RWL) as a common competitive strategy [21,22].

RWL refers to the strategy adopted by most combat sports athletes to temporally reduce their body mass, typically about 2–10%, but with reports of reductions greater than 12% [23,24], a few days before competitions to fit in a lower weight category, in an attempt to gain an advantage against lighter, smaller, and weaker opponents [22]. Achieved only through the combination of aggressive dehydration and starvation methods [24,25,26,27,28], it is a well-established common practice among combat sports athletes [29] whose harmful effects are already known and well documented in the literature [26,30,31,32,33]. RWL is not an issue restricted to adult or professional sports as children and adolescents from 10 years old also use RWL for competitive advantage [34,35]. Therefore, it was proposed that the sports community should frame RWL as doping and ban it from combat sports because of its detrimental health effects and for causing unfair competition [36].

In youth combat sports, at the same time that there is a need to verify the efficacy of body mass-based categorization to control the maturity effect providing fair competitions, reducing the injury risk and promoting engagement in sports for the long-term, it is imperative to find alternatives to body mass as a bio-banding strategy because of the RWL consequences. In this sense, among possible research designs, mediation studies can be used to understand which characteristics associated with body growth and biological maturation most strongly affect the physical performance of young athletes. Mediation analysis has recently been used in studies with children to evaluate the effect of biological maturation on motor competence performance through the mediation of anthropometric characteristics [37,38]. Therefore, the purposes of this study were: (1) to investigate whether there is an effect of biological maturation on the performance of young judo athletes after controlling chronological age and body mass; and (2) in the situation where the maturation effect is evidenced, to investigate anthropometric variables that can mediate this effect. Two hypotheses were submitted for confirmation: (1) the effect of maturation on the physical performance is significant even after age and body mass control; and (2) variables associated with body size mediate the effect of maturation on physical performance.

## 2. Materials and Methods

The participants were 67 young male judokas aged 11.0–14.7 years from eight Portuguese judo clubs. To be included in the investigation, it was necessary to have at least 12 months of judo training. Prior to data collection, parents or legal guardians signed informed consent. In addition, verbal consent was obtained from participants after the presentation of the aim and procedures. The study was approved by the Scientific Council of the Faculty of Sports Sciences and Physical Education of the University of Coimbra and was conducted in accordance with the Declaration of Helsinki for human studies of the World Medical Association.

The present study adopted common anthropometric procedures [39]. Stature and sitting height were measured using a portable stadiometer (Seca Bodymeter 206) and a segmometer (Rosscraft), respectively. The lower limb length was estimated as stature minus sitting height. Arm span was measured assessing the distance between right and left dactylion points with both arms abducted 90 degrees labeled with chalk on the wall using an anthropometric tape. The hand length was measured as the distance between the stylion and dactylion, while the foot length was measured as straight distance between the acropodion and pterion points. Arm circumference and calf circumference were measured with an anthropometric tape. All measures were taken to the nearest 0.1 cm. Body mass was measured to the nearest 0.1 kg using a portable digital scale (Seca Bella 840). Skinfold thickness was assessed to the nearest 0.1 mm using a Rosscraft skinfold calipers in the following references: triceps, subscapular, suprailiac, and calf. Estimates of fat mass percentage were obtained from the sex-specific equation derived from the sum of the triceps and subscapular skinfolds [40]. Afterward, estimated fat and fat-free masses were calculated to the nearest 0.1 kg.

The predicted mature stature (PMS) was the maturational indicator used to classify the judokas according to the maturational state. It has been used in investigations about the biological maturation effect on physical fitness of young people in general [38,41,42], in research focusing on youth performance in sport [43,44] and as a criterion for bio-banding in experimental training and tournaments [15,16,17] due to the advantages it presents compared to more valid but invasive indicators [45,46,47]. The PMS was calculated by the Khamis–Roche method [14]. The protocol requires decimal age, stature, and body mass of the participant and average parental stature. The stature of the parents was collected through questionnaire attached to the informed consent sent to the parents or legal guardians. The current stature was expressed as a percentage of PMS (%PMS). It is assumed that among children of the same chronological age, individuals closer to the PMS are more advanced in biological maturation compared with individuals who are farther [4]. From the %PMS, the z-score was calculated on the mean and standard deviation from the sample itself to classify the evaluated judokas by maturity status. Two groups contrasting in somatic maturation were derived from z-scores of attained %PMS: more mature (P > 50%) and less mature (P ≤ 50%).

The aerobic performance of the judokas was evaluated using the Pacer (Progressive Aerobic Cardiovascular Endurance Run Test) test, with the number of completed laps being used as a performance indicator. The anaerobic performance was assessed using the line-drill test, with the recording in seconds of each judokas’ time to complete the course. The agility was evaluated through the application of the 10 × 5 m shuttle-run test, recording the total time of completed laps. The following indicators were used for the assessment of the subjects’ muscle strength: abdominal muscle strength (AMS) applying the 60-s sit-ups test; lower body muscle strength (LBS) using the standing long jump test; upper body muscle strength (UBS) through the 2-kg medicine ball throw; and handgrip strength (HgS) measured by a dynamometer Lafayette model 78–10, through two attempts using the dominant hand. The best of the two attempts in kilograms was recorded.

All data were collected between April and May 2014 by the authors and a team trained by them specifically for this purpose, in a single visit where the anthropometric measurements were carried out initially, followed by physical performance evaluation. Measurements and tests were performed in circuit form. When completing all anthropometric stations, the young judoka were asked to perform warm-up exercises under the guidance of a trainee researcher and then sent to the physical performance evaluation stations in the following order: (1) Pacer; (2) 2 kg standing medicine ball throw; (3) stand broad jump test; (4) 10 × 5 m shuttle-run test; (5) sit-ups; (6) handgrip strength with a dynamometer; and (7) line-drill test.

Descriptive statistics (ranges, means, standard deviations, and 95% confidence intervals) were used for describing the anthropometric profile, the physical fitness and the maturational status of judokas in the total sample. To test normality Kolmogorov–Smirnov was used and appropriate log transformations (log 10) were adopted to normalize distributions. Pearson partial correlations were calculated between biological maturation and anthropometry and physical performances adjusting by chronological age and body mass, the common bio-banding strategy used in judo competitions. From the significant partial correlations established amid the maturational state and the physical performance variables, the physical variables that had their partial correlations with the anthropometric measurements estimated were selected. Mediation analyses aim to infer whether the relationship between a predictor and an outcome depends on the mediation of other variables. This type of analysis is based on correlation and regression tests to establish relationships among predictor, mediator, and outcome, and can be performed by different methods. In this study, mediation analyses were performed using Process (v3.3 by Andrew F. Hayes). The choice of variables to be tested as mediators and outcomes met the criterion of having a significant correlation with the predictor and correlation with each other. All linear regression models were adjusted by chronological age and body mass. Significance of *p* < 0.05 was adopted in the analyses. IBM SPSS 22.0 software (SPSS, Inc., Chicago, IL, USA) was used in the study.

## 3. Results

The descriptive statistics for the total sample and the results of normality tests are presented in Table 1. Body fat mass, agility, lower body strength, and handgrip strength tests presented significant values in the Kolmogorov–Smirnov test. Logarithmic transformation was performed in these variables for inferential statistics.

Table 2 summarizes the partial correlation coefficients among biological maturation, anthropometric, and physical performance variables. Concerning the correlation between maturity status and physical performance, only aerobic performance (*r* = 0.273, *p* < 0.05) and handgrip strength (*r* = 0.292, *p* < 0.05) presented significant correlation; therefore, they were selected to have their partial correlations with the anthropometric variables estimated. Two mediation analysis models were derived from the correlations. The first model as a predictor the maturity status, as an outcome the aerobic performance, and as variables to be tested as mediators the anthropometric variables that partial correlated simultaneously with both, namely body fat mass, body fat free mass, stature, sitting height, and inferior members length. The second model maintained the biological maturation as a predictor, but the outcome variable was handgrip strength, being tested as mediators body fat mass, body fat-free mass, stature, sitting height, arm spam, superior members length, and inferior members length.

The mediation analyses performed are shown in Figure 1. They were adjusted by chronological age and body mass, as were the correlation analyses, also considering that these variables are utilized in judo to band youth athletes in age–weight categories. In the mediation model having biological maturation as predictor and aerobic performance as an outcome, the first regression equation showed a positive total effect between them (*p* < 0.05); the second equation negatively associated maturation with body fat mass (*p* < 0.05); and the third equation, where maturation and body fat mass simultaneously participate in the model, showed a negative association between body fat mass and aerobic performance (*p* < 0.01), a loss of significance in the association between predictor and outcome (direct effect) and a significant indirect effect (95%CI [11.580–204.270]), indicating a total mediation. Total mediation was also found between maturation and aerobic performance with body fat-free mass as a mediator, given that there was a positive association between maturation and fat-free mass (*p* < 0.01), a positive association between body fat-free mass and aerobic performance (*p* < 0.01), and a not significant direct effect, whereas the indirect effect calculated had significance (95%CI [39.508–207.606]).

In Figure 1, the second group of mediation analysis itemized as predictor the biological maturation and as an outcome the handgrip strength. The first regression equation reported a positive association between maturation and handgrip strength (*p* < 0.05). The analysis of mediation resulted in five anthropometric variables exerting total mediation between the predictive and outcome variables. The second regression equations in the models indicated for biological maturation negative relationship with body fat mass (*p* < 0.05) and positive with body fat-free mass (*p* < 0.01), stature (*p* < 0.001), arm span (*p* < 0.01), and lower members length (*p* < 0.001). In all cases, the direct effects, i.e., the effect of the predictor variable on the outcome adjusted by the mediator variable, became non-significant, and the indirect effects were significant: body fat mass (95%CI [0.004–0.057]), body fat-free mass (95%CI [0.011–0.058]), stature (95%CI [0.008–0.061]), arm spam (95%CI [0.003–0.044]), and inferior members length (95%CI [0.005–0.049]). Sitting height and inferior members length, in the first model, and superior members length in the second model, are not shown in Figure 1 because their mediation effects were not found in their respective predictors and/or outcomes.

## 4. Discussion

The aim of this study was to investigate the effect of biological maturation and the role of anthropometric variables as possible mediators of this effect on young judokas physical performance, inasmuch as there is evidence that the current criteria used to grouping these athletes for training and competition, i.e., chronological age and body mass, may not be effective to control the maturation effect, but also lead to the practice of RWL. Through the partial correlations and mediation analyses, it became evident that in this sample the biological maturation still influences performance variables even with the adjustment of chronological age and body mass. The partial correlations and the two mediation models showed significant and positive maturation effect on aerobic performance and handgrip strength. The first model results exposed total mediation role of the body fat mass and body fat free mass on the maturation effect over aerobic performance, while the second model demonstrated total mediation role of these two variables, as well as of stature, arm span, and inferior members length, on the maturation effect over handgrip strength.

Aerobic performance and handgrip strength are two physical capabilities considered important for success in judo, since the sport is characterized by intermittent actions of great strength and power, interspersed with moments of recovery, where the athlete through grappling techniques seeks to throw the opponent to the ground and occasionally immobilize him/her or make him/her surrender in the ground fight [48,49]. The duration of a judo match in competitions can vary from a few seconds, when an applied technique reaches the maximum score, up to 8 min, when there is a regular 5-min tie and extra time is used to define the winner. In international competition, an athlete can perform up to seven matches. Considering the nature of motor actions, duration, and number of bouts, judo is considered a modality with both anaerobic and aerobic energy demands [50]. The aerobic system contributes to the judoka sustaining the effort for the duration of the combat and to recovering in the brief moments of effort reduction between techniques and in the rest periods between matches [51,52]. In addition to the energy demand, for pulling and pushing the opponent allowing the application of throwing, immobilizing, and blocking techniques, the development of the handgrip strength is considered critical to successful judo performance [53].

Although their objective is to equalize training routines and competitions, the distribution of young judokas and young athletes of other combat sports in age–weight categories could be inappropriate for biological maturation effect control. Previously, in an investigation regarding the relationship between chronological age and maturation effect in 54 young karateka of both sexes aged 7–16 years, maturity-advanced athletes in several weight categories were found, which would, according to the authors, lead to competitive inadequacy and evidence the need for other criteria for organizing young combat athletes [20]. As of this writing, we were unable to find further studies that investigated the existence or otherwise of the maturation effect on weight categories in young combat sports, which highlights an important gap that this study can help fill. Studies on relative age effect (RAE) in combat sports, on the other hand, are more common and show a possible RAE within weight categories, but the results of these investigations are still divergent [54,55] and, although often associated, maturational status and RAE appear to be independent [56,57]. Nevertheless, it seems appropriate to state that the use of body mass for the distribution of combat sports athletes in categories that seek equal conditions among them is affected by several factors that go beyond genetics, such as diet, lifestyle, level of physical activity, and psychological state, among others [4], even within athletes, which brings a variability that could prevent the efficient control of biological maturation, not avoiding the performance imbalance among young athletes of the same age. It is precisely the body mass possibility of manipulation that allows the adoption of the RWL strategy, where purposely and as a competitive strategy athletes undergo extreme dehydration and starvation to compete in categories with lighter opponents. The adoption of RWL only, apart from the possibility of not controlling the maturational effect, could be enough to consider weight categories as ineffective in ensuring proper training routines and fair competitions. Further, banning RWL as a competitive strategy is necessary from a public health perspective, as there are reports of combat sport athlete deaths associated with this practice [58,59].

PMS as an indicator of biological maturation has gained prominence in investigations of young athletes for its noninvasive aspect and greater practicality of application, compared with classical evaluation methods [14]. In a bio-banding experience at a young men’s soccer tournament in the UK, 66 athletes aged 11–14 years were divided into categories according to the percentage of PMS attained (85–90%) at the time of the measurement [15]. All young participants described the experience as positive and recommended that the Premier League incorporate bio-banding into their official tournaments. Furthermore, compared to chronological age-oriented tournaments, athletes advanced in maturity reported that they were more physically demanding, which required a greater emphasis on game technique and tactics, while less mature athletes reported that they enjoyed the experience the most because they had more opportunities to apply their technical and tactical knowledge as they were less physically pressured. This, according to the authors, advocates in favor of bio-banding as it allows for a holistic development of young soccer athletes. The use of PMS as bio-banding in judo and other combat sports, as in the soccer example, could be effective for controlling the maturational effect and for extinguishing the practice of RWL in these modalities, but there are no reports of experiences in this regard in the literature. However, this would not cause any change in the adoption of RWL in adult judo and combat sports. Therefore, it makes sense to search for an alternative criterion for athlete’s categorization that can be used to control maturation in combat sport for young people and to eradicate RWL in combat sport from young to adult. In fact, the search for anthropometric variables that could exert control over the maturation in youth combat sports is recurrent and several studies have been interested in seeking anthropometric variables associated with success in judo [60,61,62,63].

The results evidenced in this investigation are limited by the lack of a larger sample size and the use of indirect tests to measure physical capacity. Further, in several studies that involve the use of PMS as maturational indicator, including this one, the stature of the parents of the investigated youngsters was verified by self-reporting, a procedure that also has support in the literature [45], but might bring bias [6]. The error between predicted and actual mature stature at 18 years of age is reported to be 2.1% [14].

## 5. Conclusions

This study sought to verify if the effect of maturation is evidenced in a sample of young judokas even with control of chronological age and body mass and, if so, which anthropometric variables mediate this effect. The effect of maturation was evidenced on aerobic capacity and handgrip strength, variables considered important for success in judo. By exerting total mediation effect on these capacities, body fat mass and body fat-free mass were evidenced in relation to the aerobic capacity, and body fat mass, body fat-free mass, stature, arm span, and lower limbs’ length in relation to the handgrip strength. More investigations and intervention projects are needed to analyze bio-banding possibilities in judo and other combat sports. Attention is drawn to the search for variables with the double potential of bio-banding in youth combat sports and body mass substitution as a categorization criterion for young and adult combat athletes, aiming at the extinction of RWL as a sports strategy.

## Figures and Tables

**Figure 1 ijerph-17-00361-f001:**
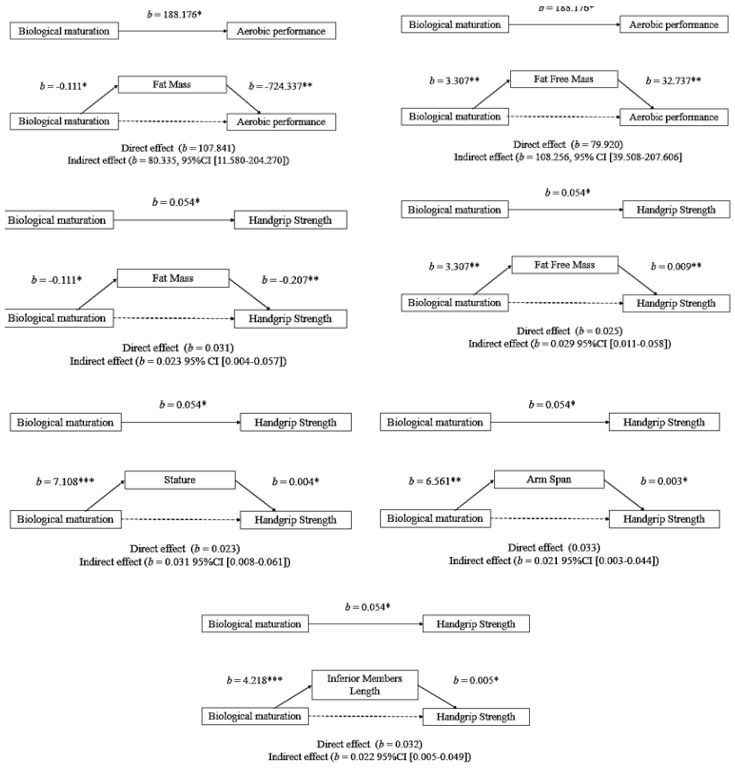
Models of body fat mass, body fat free mass, stature, arm span, and inferior members length mediation on the maturational effect on aerobic performance and handgrip strength. * *p* < 0.05; ** *p* < 0.01; *** *p* < 0.001; solid line, significant effect; dash line, non-significant effect.

**Table 1 ijerph-17-00361-t001:** Descriptive statistics for the total sample and test of normality (*n* = 67).

Variables	Range		Mean		*sd*	Kolmogorov–Smirnov	
	Minimum	Maximum	Value	95%CI		Value	*p*
Chronological age (years)	11.01	14.70	12.54	12.30–12.78	0.99	-	-
Predicted mature stature (cm)	161.9	198.3	182.6	180.2–184.3	7.2	-	-
Attained PMS (%)	77.0	94.0	84.4	83.2–85.5	4.7	-	-
Training experience (years)	1	9	3.33	2.74–3.91	2.40	-	-
Body mass (kg)	27.6	79.6	47.6	44.7–50.5	11.2	0.102	0.081
Body Fat mass (kg)	2.1	34.4	9.6	8.0–11.1	6.3	0.150	<0.01
Body Fat free mass (kg)	25.5	65.1	38.0	36.1–39.9	7.8	0.099	0.173
Stature (cm)	134.8	176.5	154.0	151.6–156.4	9.9	0.075	0.200
Sitting height (cm)	71.5	93.2	80.0	78.8–81.2	5.1	0.078	0.200
Arm span (cm)	133.0	180.0	154.1	151.5–156.7	10.8	0.060	0.200
Superior members length (cm)	36.2	70.8	60.2	58.9–61.5	5.4	0.086	0.200
Hand length (cm)	14.1	21.3	16.9	16.5–17.2	1.5	0.074	0.200
Inferior members length (cm)	60.3	85.5	74.0	72.7–75.4	5.5	0.057	0.200
Foot length (cm)	20.1	29.0	24.4	24.0–24.9	2.0	0.098	0.185
Arm circumference (cm)	19.0	36.0	25.3	24.5–26.1	3.3	0.068	0.200
Calf circumference (cm)	27.0	40.1	32.6	31.8–33.4	3.3	0.071	0.200
Pacer test (m)	140	1740	757	680–835	318	0.094	0.200
Line-drill test (s) *	30.09	46.60	36.14	35.36–36.92	3.20	0.074	0.200
Agility 10 × 5 shuttle run (s) *	15.88	26.25	19.44	18.93–19.96	2.12	0.139	<0.01
60-s sit-ups (count)	15	61	41	39–44	10	0.089	0.200
2-kg ball throw (m)	3.19	8.79	5.22	4.93–5.52	1.22	0.077	0.200
Standing long jump (m)	1.12	5.65	1.69	1.55–1.83	0.57	0.179	<0.01
Hand grip strength (kg)	14.0	40.0	24.8	23.4–26.2	5.8	0.158	<0.01

95%CI, confidence interval; *sd*, standard deviation; PMS, predicted mature stature; * in runtime tests, lower value represents better performance.

**Table 2 ijerph-17-00361-t002:** Partial correlation coefficients (controlling for chronological age and body mass) among biological maturity (given by the z-score of the attained %PMS), anthropometric, and physical variables; and partial correlation coefficients between aerobic performance and handgrip strength with anthropometric variables (*n* = 67).

Variables	Biological Maturation	Physical Fitness	
		Aerobic Performance	Handgrip Strength
Anthropometry			
Body Fat mass (kg) ^#^	−0.303 ***	−0.432 ***	−0.461 ***
Body Fat free mass (kg)	0.387 **	0.451 ***	0.453 ***
Stature (cm)	0.497 ***	0.305 *	0.395 **
Sitting height (cm)	0.400 **	0.263 *	0.322 **
Arm span (cm)	0.387 **	0.191	0.359 **
Superior members length (cm)	0.288 *	0.116	0.296 *
Hand length (cm)	0.236	0.352 **	0.247 *
Inferior members length (cm)	0.443 ***	0.258 *	0.348 **
Foot length (cm)	0.103	0.260 *	0.245
Arm circumference (cm)	−0.168	−0.040	−0.200
Calf circumference (cm)	−0.126	−0.190	−0.282
Physical fitness			
Pacer test (m)	**0.273 ***		
Line-drill test (s)	−0.225 ^##^		
Agility 10 × 5 shuttle run (s) ^#^	−0.053 ^##^		
60-s sit-ups (count)	−0.069		
2-kg ball throw (m)	0.074		
Standing long jump (m) ^#^	0.217		
Hand grip strength (kg) ^#^	**0.292 ***		

* *p* < 0.05, ** *p* < 0.01, *** *p* < 0.001; Bold – physical performance variables selected to have partial correlation with the anthropometry variables tested; ^#^ test was performed on log-transformed variable; ^##^ in runtime tests, lower value represents better performance.

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
