# Peer review of "Bio-Banding in Judo: The Mediation Role of Anthropometric Variables on the Maturation Effect"

_ijerph, 2020, doi:10.3390/ijerph17010361_

Round 1
Reviewer 1 Report
The manuscript is very interesting, and it is excellently written. I only have one question, why did you use The predicted mature stature (PMS) for the assessment of the maturity of judoka?
On the other hand, check the measurement used by the Lafayette dynamometer model 78-10, because the unit of measurement kgf is not international, could it be kg? Also correct the table 1 in red with "f".
Author Response
Please, see the attachment.

Reviewer 2 Report
The paper presented is well written and interested to read. The manuscript, "Bio-banding in judo: the mediation role of 3 anthropometric variables on the maturation effect", presents a new approach to use bio-banding based on age and body mass for judo athletes'.
The presentation of the research technique and characterisation of the results achieved indicate that the method is quite suitable and in fact could be transformative for this kind of research. However the relation of health and performance could be better explored according the journal framework. The paper is approved without amendments.
Author Response
Please, see the attachment.

Reviewer 3 Report
The study aimed to investigate whether there is an effect of biological maturation on the performance of young judo athletes after controlling chronological age and body mass; and in the situation where the maturation effect is evidenced, to investigate anthropometric variables that can mediate this effect. The manuscript is very interesting, well written and have consistent methods. I have only some minor suggestions and observation that I would like the authors to consider.
Specific comments:
Introduction:
The introduction is well structured and explains in detail the research problem by which the present investigation is carried out. Line 72-74: The sentence "Mediation analysis has recently been used in studies with children to evaluate the effect of biological maturation on motor competence performance through the mediation of anthropometric characteristics [37,38]" could be placed before the two objectives.
It would be interesting to add a hypothesis about what the authors expected to find with this study.
Materials and methods:
Line 77: The statement "and did not pretend to represent the judo athletes’ population of this country" could be delte due to in my opinion is not relevant. I think that is clear that the study does not intend to represent the entire Portuguese population. Line 84-94: The anthropometric information is very well detailed. However, would it be possible to indicate if all measurements were carried out by the same person?. Line 115: Was the manual strength test performed on the dominant hand or both? Please add this information. Line 119: Did the tests take place on the same day in all clubs? or on different days?
Results:
Results are very clear and visuals. Line 143: There is a comma in red. Table 1: The units of "Hand grip strength" are in red.Discussion:
Line 273-275: I agree with the authors since the use of parents' stature through a questionnaire is not as objective as possible. But the direct measurement of this parameter could avoid the margin of error to be reported, in many cases sure of past measurements in time.Author Response
Please, see the attachment.
